**PLOS** NEGLECTED TROPICAL DISEASES

# Identification of the area sampled by traps: A modelling study with tsetse

**Glyn A. Vale**[1,2]*, **John W. Hargrove**[1], **Steve J. Torr** [3]*

**1** DSI-NRF Centre for Epidemiological Modelling and Analysis, University of Stellenbosch, Stellenbosch, South Africa, **2** Natural Resources Institute, University of Greenwich, Chatham, United Kingdom, **3** Vector Biology Department, Liverpool School of Tropical Medicine, Liverpool, United Kingdom

* valeglyn@gmail.com (GAV); steve.torr@lstmed.ac.uk (SJT)

## Abstract

### Background

Sampling with traps provides the most common means of investigating the abundance, composition and condition of tsetse populations. It is thus important to know the size of the area from which the samples originate, but that topic is poorly understood.

### Methods and principal findings

The topic was clarified with the aid of a simple deterministic model of the mobility, births and deaths of tsetse. The model assessed how the sampled area changed according to variations in the numbers, arrangement and catching efficiency of traps deployed for different periods in a large block of homogeneous habitat subject to different levels of fly mortality. The greatest impacts on the size of the sampled area are produced by the flies' mean daily step length and the duration of trapping. There is little effect of trap type. The daily death rate of adult flies is unimportant unless tsetse control measures increase the mortality several times above the low natural rates.

### Conclusions

Formulae for predicting the probability that any given captured fly originated from various areas around the trap are produced. Using a mean daily step length ($d$) of 395m, typical of a savannah species of tsetse, any fly caught by a single trap in a 5-day trapping period could be regarded, with roughly 95% confidence, as originating from within a distance of 1.3km of the trap that is from an area of 5.3km$^2$.

## Author summary

We produced a simple, deterministic model to highlight important principles in the neglected matter of the probability that any trapped tsetse will have originated from various sizes of area around the trapping site. The modelling was kept simple by envisaging the use of just one trap, or a group of only five traps, evenly spaced inside a circular area

**Data Availability Statement:** All relevant data are within the manuscript and its Supporting information files.

**Funding:** Funding for this research was provided by the Bill and Melinda Gates Foundation (www.

gatesfoundation.org), with grants awarded to SJT (Grant ID#: OPP1104516, INV-001785), and the UK Biotechnology and Biological Sciences Research Council with grants awarded to SJT (BB/S01375X/1, BB/P005888/1, BB/L019035/1). SACEMA (GAV, JWH) receives core funding from the Department of Science and Innovation, Government of South Africa. The funders had no role in study design, data collection and analysis, decision to publish, or preparation of the manuscript.

**Competing interests:** The authors have declared that no competing interests exist.

within a large block of homogeneous habitat and operated for no more than 30 days. In deriving formulae for the sampled area, we found it appropriate to consider only the flies' mean daily step length, and the number of days of trapping. The type of trap employed was unimportant. The daily mortality of the adult tsetse population had little effect unless the death rates imposed by control measures were several times greater than the natural rates.

## Introduction

Studying the samples of tsetse (*Glossina*) caught from stationary traps is one of the most common means of producing information for the abundance, composition and condition of the tsetse population. It is crucial, therefore, to be able to interpret confidently what the catches mean about such topics. In some cases, the interpretation depends critically on understanding the size of the area sampled by the traps. Thus, De Meeûs *et al.* [1] working with a variety of tsetse species, wanted to know the sampled area as an aid to calculating population densities. In contrast, Dransfield's studies [2] with one of the tsetse species, namely *G. pallidipes*, aimed to estimate how the area sampled by one trap overlapped with areas sampled by other traps nearby, so leading to inter-trap competition and the reduction in the catch per trap. The sampled area is also important in the context of traps used to control tsetse populations, since it indicates the area over which the population is depleted [3].

Unfortunately, however, the concept of a sampled area, and the methods of establishing its size, are complicated. For example, in Dransfield's work [2] the sampled area was regarded primarily as the locality in which the flies can sense, and respond to, the stimuli from the trap at any one instant. We might call this the "proximate" area sampled. It will tend to be small because the stimuli tend not to travel far, *i.e.*, only about 10m for visual stimuli, and up to around 90m for odour attractants [4]. In contrast, the sampled area conceived by others [1, 3] is the area from which the captured flies originate in a day or more. This area, which we might call the "ultimate" sampled area, will involve more than simply the movement of flies in direct response to bait stimuli. It will also include movement at times and places in which the flies are not responding to such stimuli, either because the stimuli are not present in the place where the flies currently occur, or because the flies are in a stage of the hunger cycle not associated with host seeking. The total movement associated with the ultimate sampled area causes the flies' net average movement to be around 200-1000m per day [5, 6]. Hence, in one day of trapping the ultimate sampled area could be up to about 10,000 times as great as the proximate sampled area, and the discrepancy would be even greater if the trapping were conducted for many days. Nevertheless, given Dransfield's concept of the proximate sampled area, the way he attempted to assess it was rational. His method involved studying the way that trap catches on a single day were affected by competing traps at various short distances away, down to a few metres [2].

One way of assessing the ultimate area of sampling would be to measure how catches are affected by the presence of other traps placed at various distances away, *i.e.*, using the same principle employed in the daily studies [2], but applying it for extended periods. However, such protracted studies would be inconvenient, and complicated by seasonal variations in catch levels and wide confidence limits [2]–and, so far as we are aware, they have never been performed. De Meeûs *et al.* [1] adopted quick and simple methods of estimating the sampled area, assuming, explicitly or implicitly, that (i) the sampled area is always circular, (ii) its radius depends only on the distance between traps, and (iii) it is unaffected by the duration of

sampling, the type of habitat, and the sex and species variations in the mobilities of the flies. However, each of these assumptions seems seriously flawed and can lead to misleading predictions for the dynamics of tsetse dispersal and hence the feasibility of tsetse control [7]. Another way of assessing the sampled area involves mark, release and recapture (MRR) exercises in which flies are released at various distances from the trap, but the precision of that technique is hindered by the difficulty of getting sufficient recaptures unless the traps are far more effective than those usually employed [3].

Further problems in assessing the size of the sampled area arise from the fact that the long-term movement towards traps occurs primarily by diffusion [8]. This means that the area sampled increases as the trapping period becomes longer. It also means that the capture probability is greater for flies nearer the trap at the start of the sampling period. This has implications for the percentage of the catch that originated from within any particular distance from the trap. Viewed another way, it determines also the percentage probability that any given fly of interest came from within that distance. Regrettably, the determination of these percentages is not straightforward. For example, given that the perimeter of a circle increases with its radius, the area of the habitat far from the trap might be greater than that nearby. Hence, provided the abundance of tsetse per unit area of habitat is fairly uniform, the number of flies potentially available for capture will perhaps be greater at increased distance, thus tending to enhance to some degree the percentage of the catch that originated from far away. Against this, the farther a fly is away at the start of the sampling period, the longer it is likely to take to get near enough to perceive the trap, so the more probable it is that the fly will die before being able to contribute to the catch. Moreover, any adult flies emerging from pupae late in the sampling period will have relatively small opportunity to be caught unless they emerge close to the trap. Finally, habitat geometry can influence the mobility of the flies [9], thereby affecting the area sampled in any particular period. Putting all of these considerations together, it appears that the concept of the sampled area is complex, depending as it does not only on the sampling duration, and habitat geometry, but also on what percentage of the catch is considered.

Given the problems of defining and measuring the sampled area, it is appropriate to explore how theoretical modelling might help. Present work employs a simple model of the mobility, births and deaths of tsetse to answer some important research questions affecting sampled areas when various numbers of traps are located at different distances apart for various numbers of days. The principles exposed, using tsetse as an example, are pertinent to the sampling of other insects whose movement involves a large random component.

## Methods

### Ethics statement

There were no ethical issues since all work was theoretical.

### Model

Tsetse dispersal may be viewed as a diffusion process with the position of a fly, relative to its origin, being a normally distributed random variable [10]. This means that exact solutions can be found for simple problems, such as the calculation of the number of flies expected to arrive at a single trap—and we used them to compare the analytical results with those obtained using simulations in these cases. For more complex situations, however, analytical solutions are cumbersome and, accordingly, we restrict our view to the simulated results.

The model is available as a supplementary information item (S1 Model) and is described only briefly here. It involved a map produced in a spreadsheet of Microsoft Excel$^{(R)}$ representing a block of homogeneous habitat measuring 35.5 x 35.5km composed of 71 x 71 square cells

**Fig 1. A-D: Various arrangements of traps in an 11 x 11 cell block at the middle of a map of 71 x 71 cells.** The separation distance shown for each arrangement is the distance between a trap and its closest neighbour. That distance is half the greatest distance between traps in present arrangements. Arrows in D show what is meant in the main text by the "orthogonal" and "diagonal" directions of movement away from the central cell.

0.5km wide. All tsetse present in a cell were considered to occur in the centre of the cell when their daily movement had been completed. For such movement, 79% of the flies present in any cell at the start of a day vacated that cell deterministically to distribute evenly between the four orthogonally adjacent cells, so producing a mean daily step length of 395m. This is consistent with published data for the diffusive movement shown by a range of common species of tsetse [11]. The cell evacuation rate of 79% is the most appropriate when using orthogonal movement to represent diffusion in two dimensions over a number of days [9]. The arrangement of any traps in each quadrant of the map were always mirror images of the adjacent quadrant (Fig 1). This meant that information for the whole map could be deduced by modelling just one quadrant. Two sorts of simulation were performed separately, as below.

1. ***Mean distance moved from the origin as a function of time***. By symmetry, the mean *displacement* from the origin is always zero, but we are interested in the the random *distance*, $D$, of the particle from the origin as a function of time. Given that the size of sampled areas depends on the time course of tsetse movement, it was required to check that the model produced the patterns of movement consistent with MRR data from the field. For this check, 100 adult flies were seeded in the central cell of the map. These flies were taken as equivalent to 100 marked flies released all at once in the central cell. The subsequent diffusion of the flies was then tracked for up to 30 days, *i.e.*, not long enough to allow any seeded flies to diffuse off the map. While it is important to consider births and deaths when using MRR to assess the numbers of tsetse in a population, births and deaths are not considered when using MRR to study rates of movement [8]. Hence, births and deaths were not involved in present modelling of movement, but they had to be considered in the following work on sampled areas.

2. ***Sampled area***. For the study of the area sampled by traps, we assumed that a single trap was placed in the middle of each of one or more of the cells within a square block of 11 x 11 cells, representing 5.5 x 5.5km, located in the centre of the model's map (Fig 1). Simulations with any arrangement of traps began by seeding 100 flies in each of the cells from which the catches of any trap could possibly originate in the number of days the traps were operated. Allowing that each cell represented 0.25km$^2$, the density of the seeded population was 400 per km$^2$ of habitat. On each day of the simulated period of trapping, the flies in each cell at the start of the day were subjected to a sequence of manipulations. First, all flies due to move to adjacent cells on that day were so transferred. Next, those flies that remained in the cell, or which had just moved into it, were caught with efficiency $E$ by any trap in the cell. Flies so trapped were assigned to the cell in which they had been initially seeded. They were then removed from the map. The value of $E$ was commonly taken as 0.1. This is equivalent

to the trap catching 2.5% of the population per km$^2$ per day, which is in the range associated with the availability of *G. pallidipes* to standard types of odour-baited trap [12, 13]. However, sometimes the value of *E* was increased to 0.2, in simulation of the availability of *G. pallidipes* to traps with enhanced doses of odour [3]. On other occasions *E* was reduced to 0.02, to represent a trap with no odour [3].

After all of the above movement and trapping, a natural mortality rate, *Mnat*, and any mortality due to control, *Mcon*, were imposed on all flies remaining on the map, whether they were in their seeded cell or any other cell. The control mortality was envisaged as occurring due to devices such as insecticide-treated cattle or targets [11] which impose a roughly steady death rate that supplements natural deaths. Unless stated otherwise, *Mcon* was zero, indicating no control, and *Mnat* was 0.03 [14]. Finally, new adults emerged from pupae in each of the initially seeded cells, in numbers sufficient to ensure that the total population on the map would be stable in the absence of any trapping or control deaths. The number of emerging flies was never affected by the changes that trapping or control had on the adult breeding populations since the 30-day maximum for the trapping period was roughly equal to a pupal period, so ensuring that the emergence rate was governed by the breeding population existing prior to any imposed decline. The daily emergence of new adults per initially seeded cell was thus always 100*Mnat*. Since the simulations involved trapping or control that lasted no more than 30 days, it seemed unnecessary to allow for the density-dependent changes in birth or death rates that are commonly associated with modelling the gross changes in tsetse abundance resulting from prolonged control [15].

In interpreting the outputs of the above simulations, it was taken that the area sampled around a single trap was the territory enclosed within a circle that was centred on the trap (Fig 2A). For that circle to embrace all cells from which the captured flies originated in the trapping period, the radius of the circle had to be equal to the greatest possible movement of any fly in the trapping period, *i.e.*, the daily step length multiplied by the number of trapping days. If the sampled area was calculated in this all-embracing way, the area increased greatly as the trapping days increased. However, when there were many trapping days, very few of the captured flies came from the outer reaches of the all-embracing area. Hence, in getting a potentially more useful view of the probable origin of any particular captured fly, it was appropriate to assess the size of the various circular areas that could account for various percentages of the catch.

When several traps were deployed, the area sampled by each of the individual traps was again regarded as the territory within a circle that was centred on the trap. However, a potentially important concern when using several traps is the overall area sampled by the whole combination of the individual traps. The total area cannot then always be regarded as a simple circle, even when the traps themselves are placed in a roughly circular pattern. This is exemplified in Fig 2B which shows a situation where the traps are deployed for just a few days at distances of at least several kilometres apart, so that the many flies that started in the extensive areas between traps could not have arrived at a trap in the sampling period. Hence, it is an abstraction to regard the sampled area as a single entity, such as the area enclosed within *z* of Fig 2B. The problems get even greater when, as is common, the traps are deployed for convenience in a line along a road, as in Fig 2C. The notion of the sampled area as a single circular entity does, however, begin to make more sense when the traps are deployed in a roughly circular pattern and the trapping period is so long that there is substantial overlap in the sampled areas associated with each single trap (Fig 2D). Hence, when assessing the overall area sampled by the combinations of five traps in the present work, the deployments were always in the roughly circular patterns of Fig 1B–1D, and the trapping periods were so long as to produce substantial overlap of individual sampled areas. The overlap was deemed to be substantial

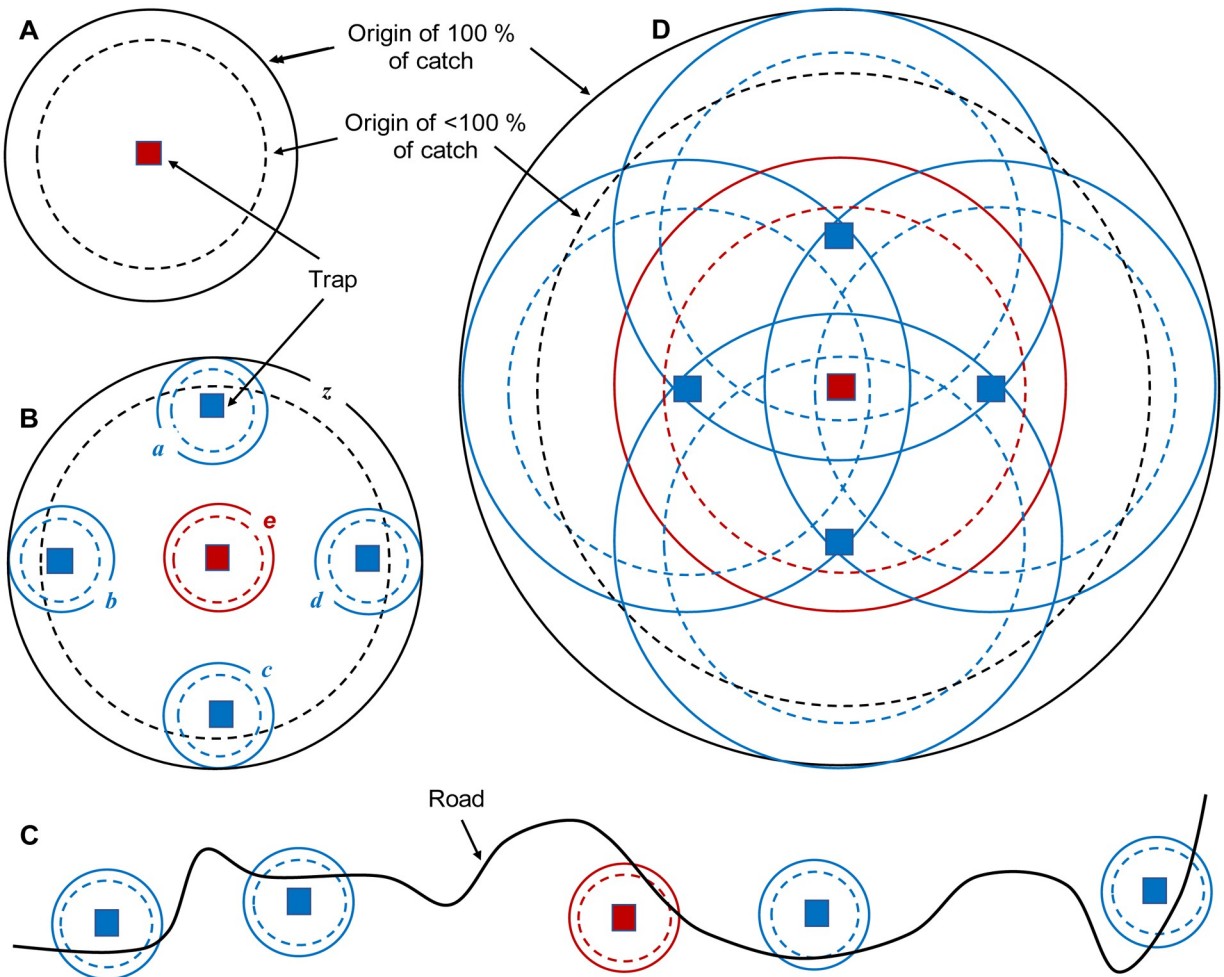

**Fig 2. Hypothetical circular areas sampled around traps in various arrangements.** A: single trap operated for several days. B: five traps placed in a roughly circular pattern but separated by a few kilometres and operated for only a few days, so that the areas sampled around each trap do not overlap; areas $z$ and $a$-$e$ are discussed in the main text. C: five traps placed in a line, separated by at least a few kilometres and operated for no more than a few days. D: five traps separated by a few kilometres and operated for many days, so that there is extensive overlap in the sampled areas. In A to D, the territory within the circles formed by solid lines indicates the areas needed to embrace the origin of all flies caught by the trap(s). Territory within dotted lines indicate smaller areas embracing the origin of only a proportion of the catch. One of the main aims of present work was to estimate the percentage of the catch originating from various sizes of such smaller areas.

when the maximum possible movement of a fly in the sampling period was greater than the separation between adjacent traps.

In calculating the percentage of total trapped flies that came from any given circular area around the trap(s), calculations were first made of the total recorded catches originating from all cells within the given circle in the trapping period. That total was then expressed as a percentage of the total recorded catches from all cells on the whole map in the trapping period that is the total catch of the trap(s).

## Results

### Mean distance moved from the origin after various days of movement

After any given number of days of movement, the way that the "marked" tsetse dispersed from the central "release" cell was closely similar in either orthogonal or diagonal directions away

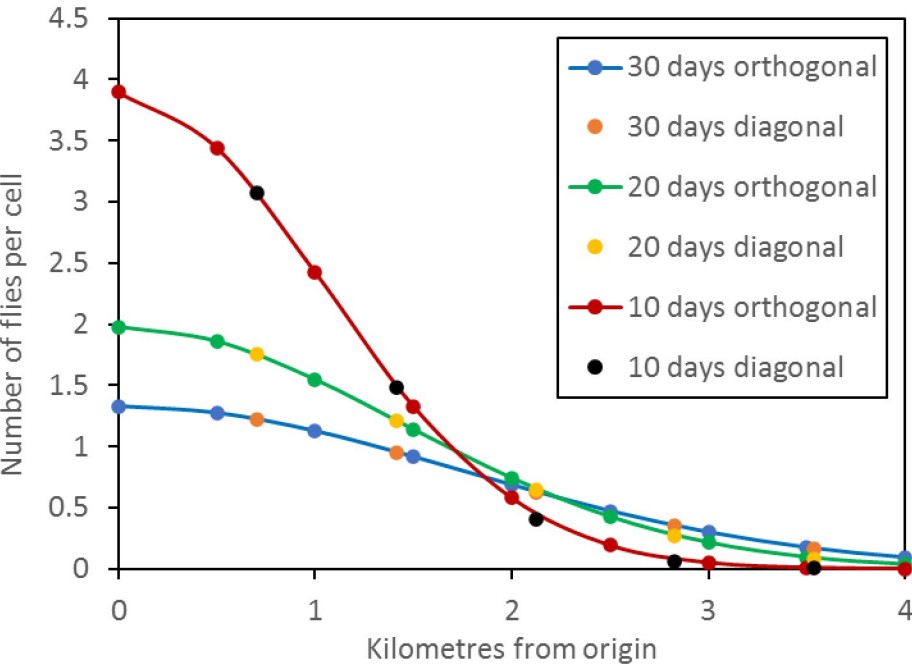

**Fig 3. Numbers of flies per cell at different distances in orthogonal and diagonal directions from the cell of origin, after various days of movement.** There were 100 flies at the origin before movement began. Fig 1D exemplifies what is meant by orthogonal and diagonal directions of movement.

from that cell. This is exemplified in Fig 3 and is as expected if the model simulated properly a two-dimensional diffusion. Notice that that the density measured on the Y-axis of Fig 3 is the number of flies per cell at various distances. To calculate the mean distance moved by the whole population it was necessary to allow for the number of cells at each distance. When that was done, the mean distance moved in any given number of days matched tolerably well the expected value derived from the accepted formula of [8]:

$$D = dt^{0.5} \qquad\qquad (1)$$

where $D$ is the mean distance from the origin after $t$ days, and $d$ is the daily step length, *i.e.*, 395m. For example, the model's simulated mean distances moved after 10, 20 and 30 days were 1.247, 1.763, and 2.159km, respectively, compared to 1.249, 1.766, and 2.164km, respectively, by Eq (1). These results and those in Fig 3 were also replicated using the approach of [10].

## Sampled area for one trap alone

As expected, the simulations showed three main effects. First, the catch on the first day was directly proportional to the catching efficiency, $E$, but declined on subsequent days because fewer flies remained to be caught. Second, the total area sampled by the trap increased with an increase in the duration of the sampling period, because the longer the period the greater the distance from which flies could arrive at the trap. Third, at any given duration, most of the total catch came from a relatively small area near the trap, with progressively fewer flies from greater distance. In consequence, the percentage of the catch that came from any given area increased rapidly with an initial increase in that area, but increased at a progressively slower

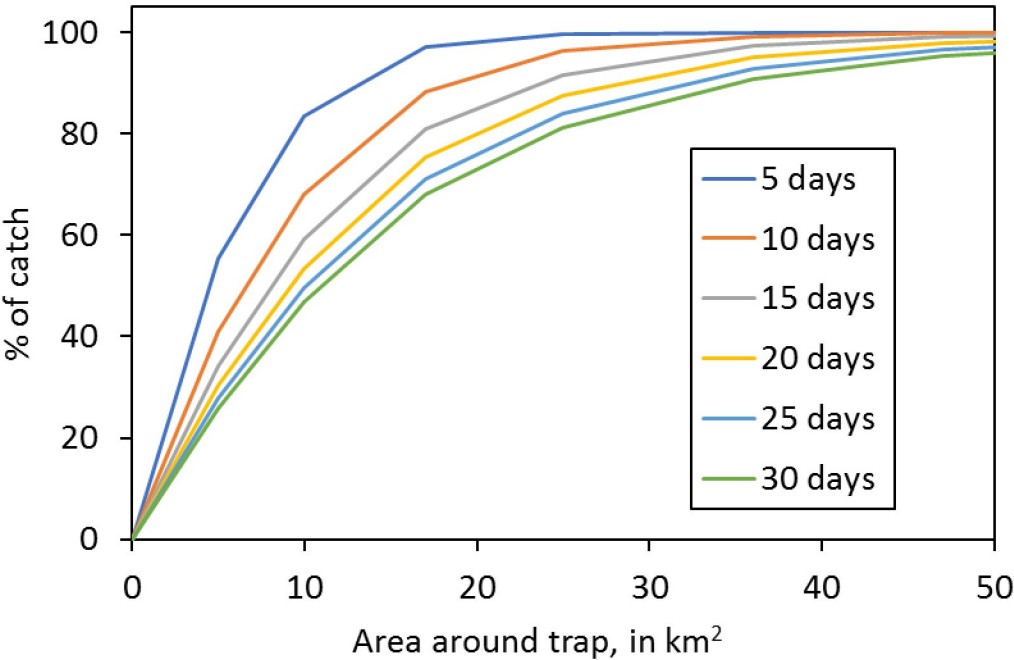

**Fig 4. Percentage of the catch originating from various circular areas around a single trap after different numbers of days of trapping.**

rate thereafter. This is illustrated in Fig 4 for a trap operating at a catching efficiency, *E*, of 0.10, when the natural mortality, *Mnat*, was 0.03 and the control mortality, *Mcon*, was zero.

Sensitivity analysis focussed on the percentage of the catch coming from two arbitrarily chosen areas close to the trap, that is within 1km and 2km of it. These areas were roughly circular, comprising 13 and 49 cells, and covering 3.25km$^2$ and 12.25km$^2$, respectively. The sampling period was five days or 30 days. The results (Table 1) showed several points of interest. First, the results confirmed the general principle (Fig 4) that the longer the trapping period the

**Table 1. The percentage of the cumulative catch originating from within given circular areas around a single trap, when various values were ascribed to the parameters *Mnat* (natural morality), *Mcon* (control morality) and *E* (catching efficiency), in five or 30 days of trapping.**

| Daily mortality | | % of catch from within 3.25km$^2$ | | | % of catch from within 12.25km$^2$ | | |
|---|---|---|---|---|---|---|---|
| *Mnat* | *Mcon* | E = 0.02 | E = 0.10 | E = 0.20 | E = 0.02 | E = 0.10 | E = 0.20 |
| 5 days of trapping | | | | | | | |
| 0.01 | 0.00 | 83.01 | 82.77 | 82.47 | 99.74 | 99.74 | 99.73 |
| 0.03 | 0.00 | 83.82 | 83.60 | 83.33 | 99.76 | 99.76 | 99.75 |
| 0.05 | 0.00 | 84.60 | 84.40 | 84.15 | 99.78 | 99.78 | 99.77 |
| 0.03 | 0.05 | 84.87 | 84.68 | 84.44 | 99.79 | 99.78 | 99.78 |
| 0.03 | 0.15 | 87.06 | 86.92 | 86.75 | 99.84 | 99.83 | 99.83 |
| 30 days of trapping | | | | | | | |
| 0.01 | 0.00 | 40.79 | 40.46 | 40.11 | 76.65 | 76.34 | 76.00 |
| 0.03 | 0.00 | 46.96 | 46.75 | 46.52 | 81.44 | 81.25 | 81.05 |
| 0.05 | 0.00 | 52.24 | 52.10 | 51.95 | 85.14 | 85.03 | 84.91 |
| 0.03 | 0.05 | 56.40 | 56.32 | 56.23 | 87.67 | 87.60 | 87.53 |
| 0.03 | 0.15 | 71.53 | 71.52 | 71.51 | 95.12 | 95.12 | 95.11 |

lower the percentage of the catch that came from near the trap. Second, the variations in *E* had very little impact on such percentages under a range of circumstances. Third, an increase in natural mortality, *Mnat*, and/or control mortality, *Mcon*, increased the percentage of the catch that originated near the trap. However, this effect was small unless the natural mortality was supplemented by very high levels of control mortality, *e.g.*, 0.15, and then only when the trapping period covered many days. The root cause of this phenomenon is the fact that daily deaths will impact most on flies coming from the furthest parts of the sampled area, because those flies take a relatively long time to reach the trap. The opportunity for this effect to become apparent increases with a rise in the death rate or a prolongation of the trapping period. Hence, when mortality increases, there is some partial counteraction to the principle that the sampled area expands with the enhanced duration of the trapping period.

**Rules of thumb.** In overview of all the single-trap simulations discussed so far, it emerged that the area within which any given percentage of the catch originates is virtually unaffected by variations in the catching efficiency, *E*. Moreover, changes in the daily death rates are of little consequence provided they are not much in excess of the low natural levels. All of this means that, in the absence of substantial control deaths, by far the greatest impact on the sampled area is the mean distance, *D*, between the start and end points of the flies' movement during the whole trapping period. The implication is that there must be rules of thumb relating such movement to the dimensions of the sampled situations.

To formulate the rules, calculations were first made of *D*, within each of the trapping periods of those simulations reported in Fig 4, that is simulations in which deaths were low and due only to natural causes, *i.e.*, M*nat* = 0.03. It was also helpful not to refer directly to the sampled areas, but rather to the radii of these areas. That meant that the origin of flies caught was regarded as being within a certain distance from the trap. The following rules emerged: 50% of the catch originates from within a radius of ~0.5*D*, 80% from within ~1.0*D*, 95% from within ~1.5*D*, and virtually 100% comes from within 2.0*D* (Fig 5).

## Sampled area with multiple traps

In these studies, the catches from all traps in any arrangement were pooled. Not surprisingly, the area from which any given percentage of the pooled catch originated was increased when the traps were spread further apart. This is illustrated in Fig 6A–6C which shows data for the various arrangements of five traps depicted in Fig 1B–1D, which involves a central trap and four others distributed symmetrically at distances of 0.5km, 1.5km or 2.5km, respectively, away from the central trap.

The red dots in Fig 6A–6C indicate the areas which are just sufficient to encompass all of the places from which the whole catch originates, as calculated by the method of De Meeûs *et al.* [1]. The method of those authors involves the assumption that the radius of the whole area sampled by three or more traps is equal to the distance between the most widely separated traps within the group. That distance is twice the inter-trap distance shown in Fig 6A–6C, as can be appreciated by considering, for example, the arrangement of traps in Fig 1D. Notice from Fig 6A–6C that the plots produced by the method of De Meeûs *et al.* [1] can be markedly erroneous as indicators of what they purport to show, *i.e.*, the areas that provide 100% of the catch. The extent of the errors depends largely on the arrangement of the traps and the duration of the trapping period.

## Discussion

The present model assumes that the habitat is uniformly good over large blocks of territory. That may be roughly true for savannah species such as *G. pallidipes*, but for species such as *G.*

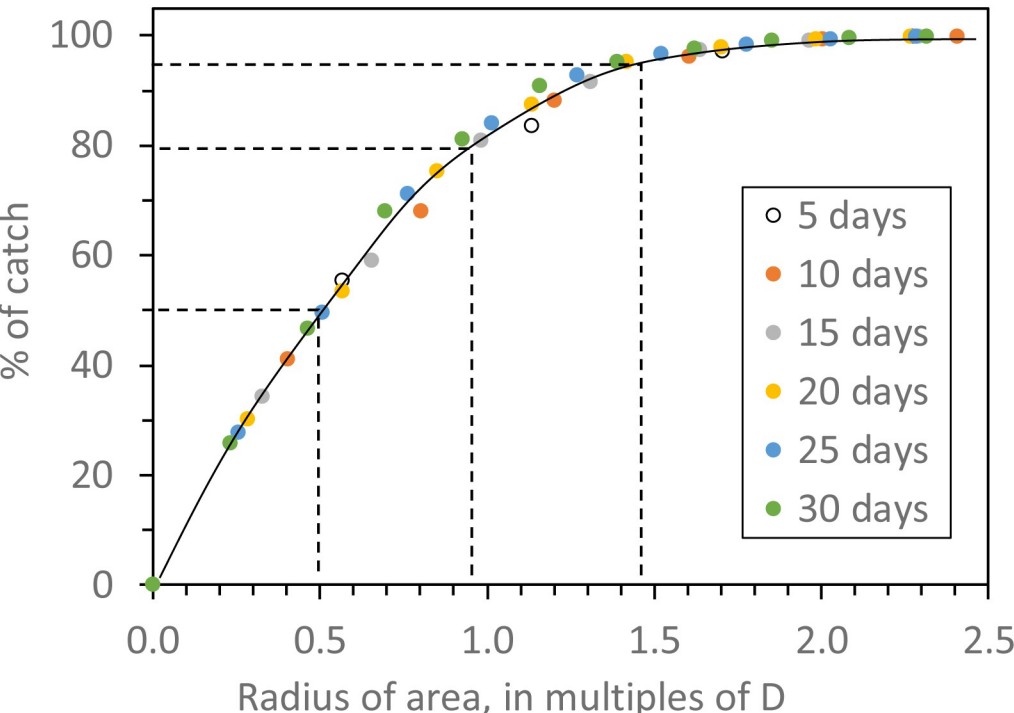

**Fig 5. Percentage of the catch originating from areas of various radii around a single trap after different numbers of days of trapping.** The radii are expressed as multiples of the mean distance *D* between the start and end points of the flies' movements during the whole duration of various trapping periods, from 5–30 days. Data are extracted from the studies of Fig 4, in which the natural daily death rate, *Mnat*, was 0.03, and the daily death rate due to control, *Mcon*, was zero, and the catching efficiency, *E*, was 0.10.

*fuscipes*, which occur mainly in woodland along rivers or lakeshores [11], the habitat is usually linear and often very patchy. Such geometry of the habitat can be expected to affect the mobility of the flies, and hence the extent of the situation sampled in any given period of trapping [6, 9]. Furthermore, no allowance has been made for variations in mobility and mortality due to

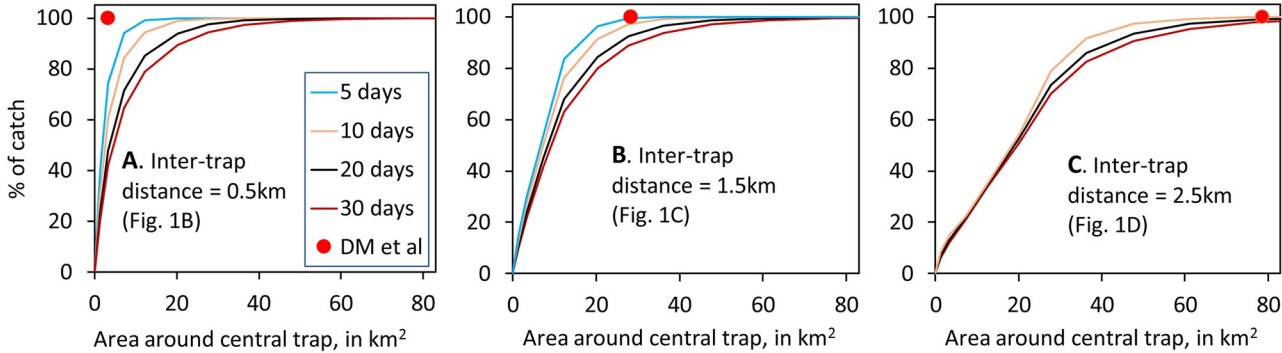

**Fig 6. Percentage of the total catch of five traps that was regarded as originating from various circular areas around the central trap, when the other traps in the group were at various distances from the central trap, as shown in the maps of Fig 1B–1D.** The inter-trap distance shown on each of the charts (A-C) of the present figure is the distance between a trap and its closest neighbour. The red plot on each chart indicates the estimated area needed to account for 100% of the catch after any number of days, as calculated by the method of De Meeûs et al. [1]. In chart C, the sampled area for five days of trapping is not shown since the separation between adjacent traps (inter-trap distance) is greater than the maximum possible distance that the flies could have moved in that time, *i.e.*, there was no substantial overlap in the situations sampled by adjacent traps. Without such overlap, the notion of a pooled area of sampling is an excessive abstraction.

the age, sex and species of the flies [5, 16] and due to season [6]. Nevertheless, present modelling is useful in highlighting some general principles that govern the size of the situation sampled, as considered below.

The sampled area varies greatly according to what percentage of the catch is considered. If the full 100% were in mind, then the area would have to be very large—in order to embrace the miniscule numbers of flies caught from extreme range. However, for practical purposes it could be appropriate to focus on areas or distances relating to lower percentages of the catch, say 95%, or even as little as 50%. Whatever percentage is adopted, the magnitude of the associated area will increase with an increase in fly mobility and the duration of the sampling period, and will decrease with a rise in the death rates of the flies. There will be no material effect of the efficiency of catching the flies arriving near the trap, so that trap design is unlikely to be of much consequence, even if it does affect the magnitude of the catch [17].

Given the quantitative relationships exposed in the present work, it can be taken that the radius of the area sampled by a single trap in homogenous two-dimensional habitat is roughly:

$$kdt^{0.5}$$

so that the sampled area is:

$$\pi(kdt^{0.5})^2$$

where $d$ is the daily step length typical of the habitat, $t$ is the days of trapping and $k$ is a constant whose value depends on what percentage of the catch is to be considered. For percentages of roughly 50%, 80%, and 95%, the values of $k$ are 0.5, 1.0, and 1.5, respectively. Thus, if the mean daily step length, $d$, were the 395m adopted in the present work, then any fly caught by a single trap in a 5-day trapping period could be regarded, with roughly 95% confidence, as originating from within a distance of 1.3km of the trap that is from an area of 5.3km$^2$. If the trapping period were 10 days, 20 days, or 30 days the distances would be 1.9km, 2.6km, and 3.2km, respectively, and the areas would be 11.3km$^2$, 21.2km$^2$, and 32.2km$^2$, respectively. If a group of traps were employed, the overall area sampled would be affected greatly by the number and arrangement of the traps. However, a rough idea of the overall area could be formed by common sense amalgamation of the sampled areas of each of the individual traps.

The above sorts of predictions are possible by using just some of the facilities offered by the model. Readers are invited to download the model and explore for themselves, and/or improve, its additional abilities to produce predictions for sampling in linear habitat, and for inter-trap competition in a range of situations. An example of the model used to consider sampling in a linear habitat is given in the supplementary materials (S1 Text).

Given the importance of tsetse mobility in the present model, and indeed in any other model of the sampling, ecology, and control of any vector, it is ideally required to have a full knowledge of the way the vector moves through the environment. For vectors as fast moving as tsetse, it is impracticable to produce the required information by direct observation. MRR studies have been helpful in some ways [5, 8] but such work indicates only the location of the fly at the times of release and recapture. What happened in between these times is unknown, especially with respect to matters such as the way that the creatures responded to contacts between vegetation types. Field experiments involving trapping associated with various placements of simulated vegetational features have offered some information of interest, but it is very crude—involving little more than showing that tsetse go round thick bushes, rather than flying through or over them[18].

The best means of elucidating movement of small fast creatures would appear to be tracking them remotely by tagging them with a device that can be followed by harmonic radar. That

system has been very successful with bees, bugs and moths [19, 20], but it remains to be applied to tsetse—that is, ironically, the insect for which it was initially developed.

## Supporting information

**S1 Text. Application of the model to sampling in a linear habitat. Table A in S1 Text**. Percentage of total catches that originated from within various lengths of sampled habitat at different durations of trapping, involving one trap operated alone, or a combination of three traps arranged at 1km intervals. The sampled lengths are centred on the position of the one trap or on the central trap of the three traps.
(DOCX)

**S1 Model. A programme to explore the concept of the size of the area sampled by tsetse traps.**
(XLSM)

## Author Contributions

**Conceptualization:** Glyn A. Vale, John W. Hargrove, Steve J. Torr.

**Data curation:** Glyn A. Vale.

**Formal analysis:** Glyn A. Vale, John W. Hargrove.

**Funding acquisition:** Steve J. Torr.

**Methodology:** Glyn A. Vale.

**Project administration:** Steve J. Torr.

**Software:** Glyn A. Vale.

**Visualization:** Glyn A. Vale.

**Writing – original draft:** Glyn A. Vale, John W. Hargrove, Steve J. Torr.

**Writing – review & editing:** Glyn A. Vale, John W. Hargrove, Steve J. Torr.

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
