## [Decision Letter · Decision Letter 0]

25 Nov 2022

Dear Dr. Torr,

Thank you very much for submitting your manuscript "Identification of the area sampled by traps: a modelling study with tsetse" for consideration at PLOS Neglected Tropical Diseases. As with all papers reviewed by the journal, your manuscript was reviewed by members of the editorial board and by several independent reviewers. The reviewers appreciated the attention to an important topic. Based on the reviews, we are likely to accept this manuscript for publication, providing that you modify the manuscript according to the review recommendations. 

Sincerely,

Adly M.M. Abd-Alla, Prof asso.

Academic Editor

Esther Schnettler

Section Editor

Reviewer's Responses to Questions

**Key Review Criteria Required for Acceptance?**

**Methods**

-Are the objectives of the study clearly articulated with a clear testable hypothesis stated?

-Is the study design appropriate to address the stated objectives?

-Is the population clearly described and appropriate for the hypothesis being tested?

-Is the sample size sufficient to ensure adequate power to address the hypothesis being tested?

-Were correct statistical analysis used to support conclusions?

-Are there concerns about ethical or regulatory requirements being met?

Reviewer #1: The design is appropriate, the parameters are clearly stated and defined, and the analysis is appropriate.

There are no ethical concerns as this is a purely theoretical study.

Reviewer #2: Accept

**Results**

-Does the analysis presented match the analysis plan?

-Are the results clearly and completely presented?

-Are the figures (Tables, Images) of sufficient quality for clarity?

Reviewer #1: The analysis matches the plan, the results are clearly presented and the figures and tables are of sufficient quality, with the exception that hte table headings have been truncated and need to be restored.

Reviewer #2: Accept

**Conclusions**

-Are the conclusions supported by the data presented?

-Are the limitations of analysis clearly described?

-Do the authors discuss how these data can be helpful to advance our understanding of the topic under study?

-Is public health relevance addressed?

Reviewer #1: The conclusions are well supported, the limitations are described and discussed, and the utility of the model is presented. The public health significance is adequately covered in the introduction.

Reviewer #2: Accept

**Editorial and Data Presentation Modifications?**

Reviewer #1: There are no significant changes required. The text needs to be revised to remove a few minor drafting errors, such as repeated prepositions and missing articles. Table 1 needs to be reformatted to make the headings visible.

Reviewer #2: Accept

**Summary and General Comments**

Reviewer #1: The authors present a deterministic model of tsetse trapping probability with time and distance from traps. They are able to simplify the model such that only the mean daily dispersal distance affects the outcome for a range of reasonable values of the other parameters. The authors assume a near uniform habitat for the two dimensional dispersion, but discuss the limitations of this and provide additional information on a linear situation as encountered with riverine tsetse species. The model used is available as supplementary material, allowing the readers to experiment for themselves on the effect of changing the parameter values in different situations.

The manuscript is clearly written and can be accepted for publication without any revision. The figures are clear and relevant to the work.

Reviewer #2: This is an interesting paper dealing with a theoretical study in the form of a model to elucidate the area of habitat that a tsetse trap can sample, allowing for daily tsetse fly movement, mortality and births. The results illustrate the effect of daily step lengths and duration of trapping on numbers of flies collected and that a trap can sample an area of roughly 5.5km2. Notably, the trap type does not have a large impact on numbers of flies collected. 

This information is useful for field studies of tsetse flies and contributes to the knowledge and efficiency of trap placement and duration of trapping.

PLOS authors have the option to publish the peer review history of their article (what does this mean?). If published, this will include your full peer review and any attached files.

Reviewer #1: Yes: Andrew Parker

Reviewer #2: Yes: Johan Esterhuizen

Figure Files:

Data Requirements:

Reproducibility:

References

---

## [Editor Report · Decision Letter 1]

6 Jan 2023

Dear Dr. Torr,

We are pleased to inform you that your manuscript 'Identification of the area sampled by traps: a modelling study with tsetse' has been provisionally accepted for publication in PLOS Neglected Tropical Diseases.

Best regards,

Adly M.M. Abd-Alla, Prof asso.

Academic Editor

Esther Schnettler

Section Editor

---

## [Editor Report · Acceptance letter]

24 Jan 2023

Dear Dr. Torr,

We are delighted to inform you that your manuscript, "Identification of the area sampled by traps: a modelling study with tsetse," has been formally accepted for publication in PLOS Neglected Tropical Diseases.

Best regards,

Shaden Kamhawi

co-Editor-in-Chief

Paul Brindley

co-Editor-in-Chief
